# Prevalence of type-2 diabetes and prediabetes in Malaysia: A systematic review and meta-analysis

**Sohail Akhtar**[1]*, **Jamal Abdul Nasir**[2], **Aqsa Ali**[2], **Mubeen Asghar**[2], **Rizwana Majeed**[2], **Aqsa Sarwar**[2]

1 Department of Mathematics and Statistics, The University of Haripur, Haripur, Khyber Pakhtunkhwa, Pakistan, 2 Department of Statistics, GC University Lahore, Lahore, Pakistan

* s.akhtar@uoh.edu.pk, Akhtar013@gmail.com

**Data Availability Statement:** All the data is inside in the paper.

**Funding:** The authors received no specific funding for this work.

## Abstract

### Objective

The main purpose of this study was to investigate the pooled prevalence of prediabetes and type-2 diabetes in the general population of Malaysia.

### Method

We systematically searched Medline (PubMed), Embase, Web of Science, Google Scholar and Malaysian Journals Online to identify relevant studies published between January 1, 1995, and November 30, 2021, on the prevalence of type-2 diabetes in Malaysia. Random-effects meta-analyses were used to obtain the pooled prevalence of diabetes and prediabetes. Subgroup analyses also used to analyze to the potential sources of heterogeneity. Meta- regression was carried to assess associations between study characteristics and diabetes prevalence. Three independent authors selected studies and conducted the quality assessment. The quality of the final evidence was evaluated using the Grading of Recommendations Assessment, Development and Evaluation (GRADE) approach.

### Results

Of 2689 potentially relevant studies, 786 titles and abstract were screened. Fifteen studies with 103063 individuals were eligible to be included in the meta-analyses. The pooled prevalence of diabetes was 14.39% (95% CI, 12.51%–16.38%; $I^2$ = 98.4%, 103063 participants from 15 studies). The pooled prevalence of prediabetes was 11.62% (95% CI, 7.17%–16.97%; $I^2$ = 99.8, 88702 participants from 9 studies). The subgroup analysis showed statistically significant differences in diabetes prevalence by the ethical sub-populations with highest in Indians (25.10%; 95% CI, 20.19%–30.35%), followed by Malays (15.25%; 95% CI, 11.59%–19.29%), Chinese (12.87%; 95% CI, 9.73%–16.37%), Bumiputeras (8.62%; 95% CI, 5.41%–12.47%) and others (6.91%; 95% CI, 5.71%–8.19%). There was no evidence of publication bias, although heterogeneity was high ($I^2$ ranged from 0.00% to 99·8%). The quality of evidence based on GRADE was low.

**Competing interests:** The authors have declared that no competing interests exist.

## Conclusions

Results of this study suggest that a high prevalence of prediabetes and diabetes in Malaysia. The diabetes prevalence is associated with time period and increasing age. The Malaysian government should develop a comprehensive approach and strategy to enhance diabetes awareness, control, prevention, and treatment.

## Trial registration

Trial registration no. PROSPERO CRD42021255894; https://clinicaltrials.gov/.

## Introduction

Diabetes mellitus is one of the most serious worldwide public health issues, posing a significant global burden on both public health and socioeconomic development. Although the incidence of diabetes has begun to decline in some nations, diabetes prevalence has climbed in most other developing and developed countries in recent decades [1–6]. According to the International Diabetes Federation (IDF), 9.3 percent (463 million) of adults worldwide have diabetes in 2019. The number is expected to rise to 10.2% (578 million) by 2030 and 10.9% (700 million) by 2045 if effective prevention methods are not implemented [7,8]. Furthermore, in 2017, nearly half of all people with diabetes (50.1%) were undiagnosed, approximately 374 million individuals (18–99 years) [8]. Similarly, prediabetes is estimated to affect 374 million (7.5%) of the global population in 2019 and is expected to increase to 8.0 percent (454 million) by 2030 and 548 million (8.6%) by 2045, with 48.1% of individual with prediabetes are under the age of 50 [8]. Type-2 diabetes reduces the average lifespan by around ten years [9].

Malaysia has the highest rate of diabetes in Western Pacific region and one of the highest in the world and costing around 600 million US dollars per year [10,11]. The prevalence of diabetes raised from 11.2% in 2011 to 18.3% in 2019, with a 68.3% increase [12]. According to a national survey report, in Malaysia in 2019, 3.6 million adults (18 and above years) had diabetes, 49% (3.7 million) cases were undiagnosed [13]. Diabetes is expected to affect 7 million Malaysian adults aged 18 and older by 2025, posing a major public health risk with a diabetes prevalence of 31.3% [12]. The prevalence of diabetes in Malaysia, based on published articles, ranges from 7.3% to 23.8% [14,15]. The increasing trend is a result of a variety of causes, including population expansion, population ageing, urbanization, and rising rates of obesity and physical inactivity [16]. The alarming prevalence of diabetes and its complications in Malaysia prompted this study to systematically identify, summarize available evidence on the prevalence of diabetes and prediabetes, and to estimate the pooled prevalence of diabetes and prediabetes in Malaysia. To our knowledge, no prior effort has been made to combine existing data on the prevalence of diabetes and prediabetes in Malaysian populations.

## Methods

### Design and registration

Our systematic review and meta-analysis protocol was registered with PROSPERO in March 2021 (registration number CRD42021255894). We conducted this study in accordance with the PRISMA guidelines [17], and the PRISMA 2009 checklist is attached in supplementary file (S1 File).

## Literature search

Similar to our previous systematic reviews [18–20], we systematically searched Medline (PubMed), Web of Science, Google scholar, Embase and Malaysian Journals Online to identify relevant studies published between January 1, 1995, and November 30, 2021, on the prevalence of prediabetes and diabetes in Malaysia. The following keywords were combined to design the search strategy: "diabetes", "type-II diabetes", "type 2 diabetes", "prediabetes", "T2D", "non-communicable diseases", "prevalence", "impaired fasting glucose", "impaired glucose tolerance", "risk factor", "risk factors", "epidemiology", "glucose abnormalities", "glucose intolerance", "Malaysia", "Malaysian" and "Malays", as well as variations thereof. To identify potential additional studies and reports, we also scrutinized the reference list of all selected articles. The results of search strategy are provided in the supplementary file (S1 Table).

## Inclusion and exclusion criteria

For this study, studies were included if they provided enough data to calculate prevalence of diabetes and prediabetes; included a community-or population-based survey and published in English between January 1995 to September 2021; participants residing in Malaysia. The following studies were excluded: were review articles, case studies, qualitative studies, case series, abstracts, and intervention studies; was irrelevant to type-2 diabetes; reported on gestational or type-1 diabetes; based on Malaysian community living outside of Malaysia; and based on duplicated information (data); and based on data that was published in more than one study (the latest data version was considered).

## Data extraction

Three investigators (JAN, AA, and RM) independently conducted data extraction using a pre-conceived and standardized data extraction form. The collected information was: surname name of the author, year of publication, year of investigation, study design, state where study was conducted, mean or median age of participants, total sample size, percentage of male participants, percentage of hypertensive participants, sampling strategy, percentage of smoker participants, area (rural vs urban), diagnostic criteria and percentage of overweight or obese participants. Disagreements and uncertainties were addressed by mutually consensus or consultation with the 4th investigator (SA).

## Methodological quality assessment

Three of investigators (JAN, AA, and RM) independently evaluated the quality of each study by using the JBI Critical Appraisal Checklist for Studies Reporting Prevalence Data [21]. Any discrepancies were discussed between the investigators, and another investigator (SA) made a final judgment if no mutual consensus could be reached. Each study received a score ranging from 0 to 9. We classified each study as high risk (for scores <50%), moderate risk (for quality scores above 50–69%), or low risk of bias based on its score (for quality scores ≥70%).

## Statistical analysis

Statistical software R (version 4.1.0) was used to generate meta-analysis for diabetes and prediabetes prevalence. Because of the significant expected heterogeneity among studies, random-effects meta-analyses were used. Prior to pooling prevalence estimates, the Freeman-Tukey double arc-sine transformation was applied to stabilise the variance of the raw prevalence estimates from each included study. The $I^2$ index was considered to evaluate heterogeneity between studies, with $I^2$ values between 25% or below indicate mild degree of heterogeneity,

between 26%-50% indicate moderate degree of heterogeneity, and greater than 50% indicate the presence of substantial heterogeneity [22–24]. Forest plots were created to visually assess the results of pooling.

We used subgroup meta-analysis to explore causes of substantial heterogeneity. For this purpose, we stratified meta-analyses by area of residence (urban vs rural), participant age group, gender, time period, and ethical subgroup. We investigated sources of heterogeneity with a meta-regression. The covariates in the meta-regression considered were state where study was conducted, study year, area of residence (rural vs urban), sample size, baseline year of data collection, mean age of participants, methodological quality, and gender. Visual inspection of the funnel plots was performed to evaluate publication bias in the meta-analyses. Egger's test and Begg's test method were also considered for quantitative estimate of the publication bias. Cohen's coefficient was used to determine inter-rater agreement among the authors who were engaged in data extraction and study selection [25]. The quality of this systematic review and meta-analysis was evaluated using the Grading of Recommendations Assessment, Development and Evaluation (GRADE) approach [26].

## Results

The PRISMA diagram with the flow of articles through this study is described in Fig 1. Our initial electronic search identified 2682 potentially relevant articles, and an additional 7 articles were identified by checking reference lists. After excluding of duplicates (n = 1903), we screened 786 articles by title and abstract and excluded 733 irrelevant articles. Inter-rater agreement among the investigators on abstract selection was very high (κ = 0.846, p<0.001). We scrutinized the eligibility of potentially relevant studies (n = 53) in full text against the inclusion and exclusion criteria. The articles were removed as 12 studies did not mention the results of diabetic patients, duplication of dataset (n = 10), only assessed patients with type-1 diabetes (n = 11), did not include enough information to compute prevalence (n = 5). Thus, 15 studies were identified as the eligible studies in the meta-analysis (Fig 1).

Table 1 summarizes the general characteristics of 15 selected studies [11,14,15,27–38]. In total, 103063 individuals were included. A cross-sectional study design was utilised in 11 of the 15 investigations, and 4 studies did not explicitly describe a study strategy, it can be inferred as a cross-sectional study design. Sample sizes ranged from 119 to 34539 individuals, with a median of 1489 individuals. The age of participants enrolled ranged from 15 to 85 years overall in 15 studies. The studies were published between 1996 and 2020, while the participants were enrolled between November 1993 and June 2019. In 14 studies, the proportion of male participants in the study sample was recorded. Male participation ranged from 0 to 54.8%, and the percentage of overweight participants ranged from 11 percent to 49.1 percent. Following an evaluation of the studies' quality, 11 were found to be of low-risk bias, 5 to be of moderate level, and none to be of high-risk bias. The authors had a high level of agreement on the extracted data (k = 0.86, p = 0.001).

### Quantitative synthesis

The findings of the overall and subgroup meta-analyses are shown in Table 2. The diabetes prevalence was reported in 15 studies, with a total of 103063 individuals. The prevalence of diabetes ranged from 7.8% to 23.8%. The overall pooled estimated prevalence of diabetes was 14.39% (95% CI, 12.51%–16.38%; prediction interval: 7.37–23.23). Fig 2 shows diabetes prevalence estimates derived from meta-analysis. The analysis revealed a significant heterogeneity. The level of heterogeneity in the meta-analysis was high ($I^2$ = 98.4%; P < 0.001) (Fig 2). The Egger's and Begg's tests did not find any publication bias in the meta-analysis (p = 0.7296 and

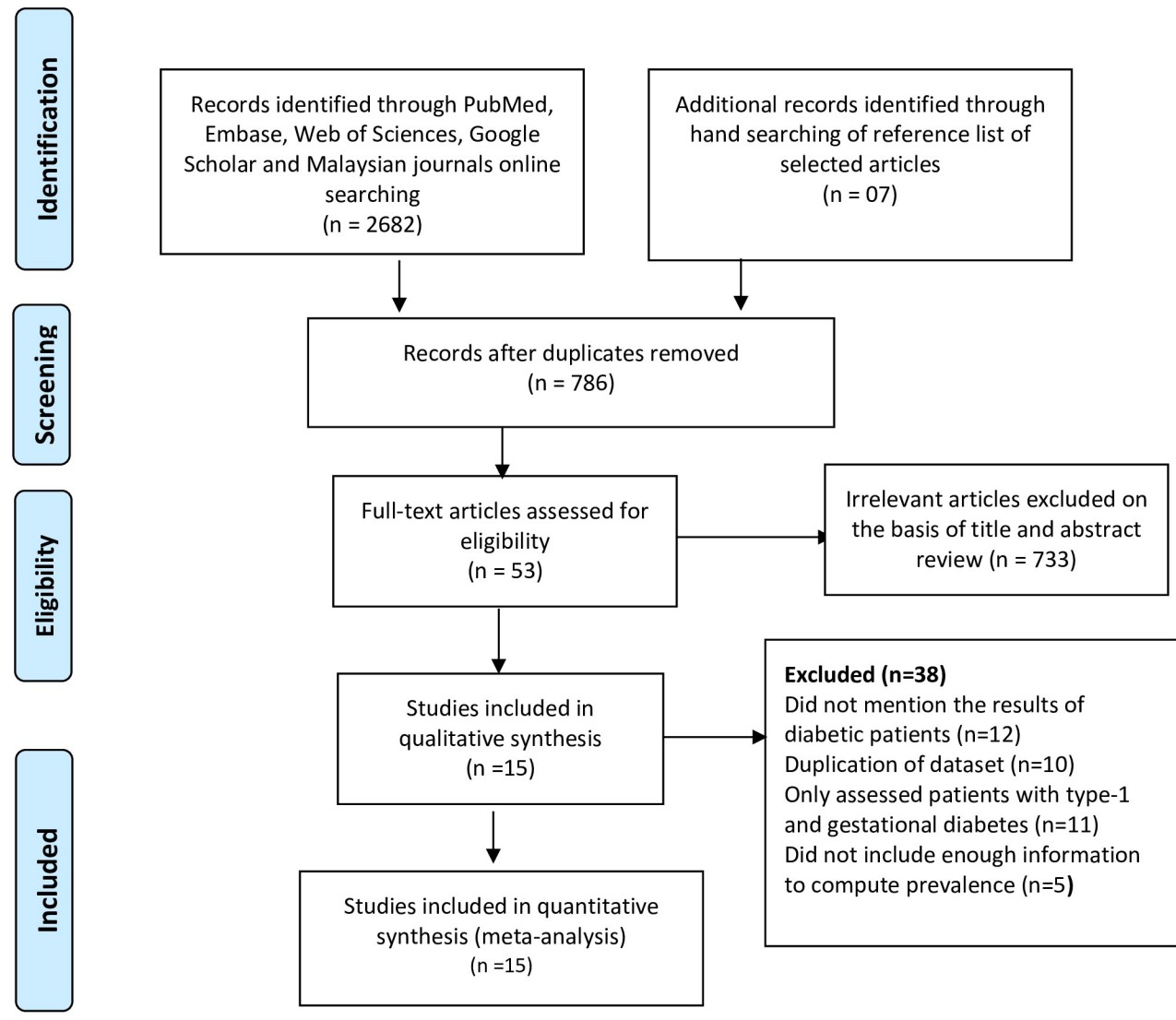

**Fig 1. Study and data inclusion, using PRISMA 2009 guideline [17].**

p = 0.5862, respectively). The visual inspection of funnel plot also showed no evidence of the presence of publication bias (Fig 3). The sensitivity analysis revealed that the pooled prevalence of diabetes varied from 13.76% (95% CI, 11.90%–15.72%) to 14.91% (95% CI, 12.96%–16.98%) after removing a one study at one time (Supplement file, S1 Fig), but no single study had a significant influence on the pooled prevalence.

The prevalence of prediabetes was reported in 9 studies, with a total of 88702 individuals. The overall pooled estimated prevalence of prediabetes was 11.62% (95% CI, 7.17%–16.97%, prediction interval: 0.25–35.23) by random-effects meta-analysis. The meta-analysis had a high level of heterogeneity ($I^2$ = 99.8%; P <0.001) (Fig 4). The visual inspection of the funnel plot (Fig 5) suggested no evidence of the presence of publication bias in the analysis. This was also not found to be statistically significant by the Egger's test of bias. The sensitivity analysis revealed that the pooled prevalence of diabetes varied from 10.43% (95% CI, 6.56%–15.07%) to 12.84% (95% CI, 7.40%–19.49%) after removing a one study at one time (Supplement file S2 Fig), but no single study had a significant influence on the pooled prevalence of prediabetes.

**Table 1. Summary of the general characteristics selected studies.**

| Author | Year | Data collection Year | Sample Size | Positive for type-2 diabetes | Positive for prediabetes | Prevalence of diabetes | Avg. Age of participant | Research Design | Setting | Male (%) | State | Sampling strategy | Diagnostic method And criteria | % of hypertension | % of Over-weight/ Obese | % of smoker | Risk Bias |
|---|---|---|---|---|---|---|---|---|---|---|---|---|---|---|---|---|---|
| NHMS-2019 [11] | 2019 | 2019 | 10464 | 1915 | NA | 18.3 | NA | NA | Both | 49.9 | National | Two Stage Stratified Random | FBG≥7 mmol/l, 2hBG>11.1 mmol/l | 32.7 | NA | 11.5 | High |
| Harris et al. [14] | 2019 | February to May 2015 | 330 | 24 | NA | 7.3 | 43.7 | CS | Rural | 40.3 | Eastern Sabah | Stratified Random Sampling | FBG≥7 mmol/l | 26.79 | NA | 18.45 | Medium |
| Samsudin et al. [15] | 2016 | September 2012 to February 2013 | 1414 | 337 | NA | 23.8 | 69.31 | CS | Urban | NA | Northern Areas | Stratified, Multistage, and Snowball | NA | 42.4 | NA | NA | Medium |
| Khebir et al. [27] | 1996 | November 1993 to January 1994 | 260 | 38 | 30 | 14.6 | 46.5 | CS | Rural | 43.1 | Kuala Selangor | Simple Random sampling | 2hPG>11.1 mmol/l | NA | 11 | NA | High |
| NHMS-2006 [28] | 2006 | 2006 | 34539 | 4007 | 1451 | 11.6 | NA | CS | Both | 44.8 | National | Two Stage Stratified Random | FPG≥7 mmol/l, 2hPG≥11.1 mmol/l | NA | NA | NA | High |
| Nazri at al. [29] | 2008 | September 2005 | 348 | 27 | NA | 7.8 | 40.7 | CS | Rural | 47.3 | Pulau Kundur | Simple Random Sampling | NA | 12.6 | 49.1 | 55.8 | Medium |
| Rampal at al. [30] | 2009 | 2004 | 7683 | 1168 | 1321 | 15.2 | 46.8 | CS | Urban | 39.6 | National | Stratified Two Stage Cluster Sampling | FBG≥7 mmol/l | NA | NA | NA | High |
| Mohamud et al. [31] | 2010 | NA | 119 | 10 | 20 | 8.5 | 35.83 | NA | Rural | 0 | Peninsular | NA | FPG≥7 mmol/l | NA | NA | NA | High |
| NHMS-2011 [32] | 2011 | 2011 | 17783 | 2703 | 871 | 15.2 | NA | NA | Both | 46.7 | National | NA | FBG≥6.1 mmol/l, 2hBG>11 mmol/l | 32.7 | NA | NA | High |
| Mustafa et al. [33] | 2011 | April to August 2008 | 3879 | 489 | 857 | 12.6 | 50.93 | CS | Both | 34 | National | NA | FPG≥7 mmol/l, 2hPG≥11.1 mmol/l | 76.5 | NA | 10.8 | High |
| Nazaimoon et al. [34] | 2013 | NA | 4336 | 993 | 939 | 22.9 | 48.02 | CS | Both | 35.1 | National | Two Stage Stratified Random | FPG≥7 mmol/l, 2hPG≥11.1 mmol/ l,6.3≤HbA1C≤6.5 mmol/l | NA | NA | NA | High |
| NHMS-2015 [35] | 2015 | 2015 | 19935 | 3489 | 937 | 17.5 | NA | NA | Both | 47.6 | National | NA | FBG≥6.1 mmol/l, 2hBG>11 mmol/l | 30.3 | NA | NA | High |
| Aniza et al. [36] | 2016 | March to November 2011 | 1489 | 162 | NA | 10.9 | 44.9 | CS | Rural | 38.7 | Tanjung Karang | Convenient Sampling | NA | 20.6 | NA | NA | High |
| Naggar et al. [37] | 2017 | March 2016 | 316 | 34 | NA | 10.8 | NA | CS | NA | 53.5 | Selangor | Simple Random Sampling | NA | 30 | 19.7 | 21.3 | Medium |
| Rahim et al. [38] | 2020 | August to November 2017 | 168 | 33 | 17 | 19.6 | 52.63 | CS | Rural | 54.8 | Penang | Simple Random Sampling, Convenience Sampling | NA | 39.5 | 23.7 | 25.2 | Medium |

Abbreviations; HbA1C, glycated hemoglobin; FBG, fasting blood glucose, FBG, fasting blood glucose, FPG, fasting plasma glucose CS, cross-sectional; 2hBG, 2-hour blood oral glucose; mmol/l, millimoles per liter, NA, not available.

**Table 2. Summary of overall and subgroup meta-analyses.**

| Variable | Studies | Sample size | Positive cases | Prevalence, % (95% CI) | I² | 95%, Prediction interval | p-Heterogeneity | p Egger | P-Difference |
|---|---|---|---|---|---|---|---|---|---|
| Prediabetes | 9 | 88702 | 6443 | 11.62 (7.17–16.97) | 0.998 | (0.29–35.23) | < 0.001 | 0.1209 | 0.8978 |
| Male prediabetes | 7 | 39241 | 2796 | 10.98 (6.95–15.79) | 0.994 | (0.58–31.47) | < 0.001 | | |
| Female prediabetes | 8 | 49201 | 3565 | 11.40 (6.55–17.40) | 0.997 | (0.01–37.55) | < 0.001 | | |
| Diabetes | 15 | 103063 | 15429 | 14.39 (12.51–16.38) | 0.984 | (7.37–23.23) | < 0.001 | 0.7296 | |
| Undiagnosed | 8 | 91526 | 6798 | 8.60 (6.48–10.99) | 0.992 | (2.25–18.48) | < 0.001 | 0.2363 | |
| By Gender | | | | | | | | 0.7555 | 0.6063 |
| Male | 12 | 45580 | 6697 | 13.80 (11.94–15.77) | 0.961 | (7.53–21.56) | < 0.001 | | |
| Female | 13 | 55678 | 7992 | 14.54 (12.50–16.70) | 0.973 | (7.49–23.40) | < 0.001 | | |
| By Setting | | | | | | | | 0.8733 | 0.0594 |
| Rural | 11 | 37307 | 5060 | 12.72 (10.63–14.97) | 0.966 | (5.84–21.73) | < 0.001 | | |
| Urban | 7 | 61561 | 9285 | 15.89 (13.59–18.34) | 0.984 | (8.23–25.48) | < 0.001 | | |
| By Age | | | | | | | | 0.3720 | 0.0001 |
| 20 to 29 | 5 | 2078 | 110 | 3.16 (3.62–6.94) | 0.493 | (1.35–11.05) | 0.0915 | | |
| 30 to 45 | 4 | 5935 | 815 | 13.71 (12.85–14.60) | 0.000 | (11.84–15.69) | < 0.001 | | |
| 46 to 59 | 4 | 11165 | 2517 | 25.66 (20.60–31.07) | 0.966 | (5.70–53.57) | < 0.001 | | |
| 60+ | 6 | 10191 | 3489 | 33.45 (28.45–38.64) | 0.955 | (17.14–52.08) | < 0.001 | | |
| Time period | | | | | | | | 0.7296 | 0.0210 |
| 1995–2010 | 5 | 42949 | 5250 | 11.82 (9.44–14.43) | 0.951 | (4.56–21.81) | < 0.001 | | |
| 2011–2020 | 10 | 64615 | 11003 | 15.77 (13.75–17.89) | 0.974 | (8.98–24.00) | < 0.001 | | |
| Ethnicity | | | | | | | | 0.1269 | <0.0001 |
| Malay | 10 | 56435 | 7718 | 15.25 (11.59–19.29) | 0.993 | (3.70–32.67) | 0.0001 | | |
| Chinese | 9 | 18057 | 1949 | 12.87 (9.73–16.37) | 0.974 | (3.29–27.38) | 0.0233 | | |
| Indian | 9 | 7909 | 1724 | 25.10 (20.19–30.35) | 0.959 | (9.14–45.65) | 0.0001 | | |
| Bumiputeras | 9 | 9699 | 704 | 8.62 (5.41–12.47) | 0.968 | (0.37–25.27) | 0.3535 | | |
| Others | 6 | 1710 | 122 | 6.91(5.71–8.19) | 0.000 | (5.25–8.76) | 0.8290 | | |

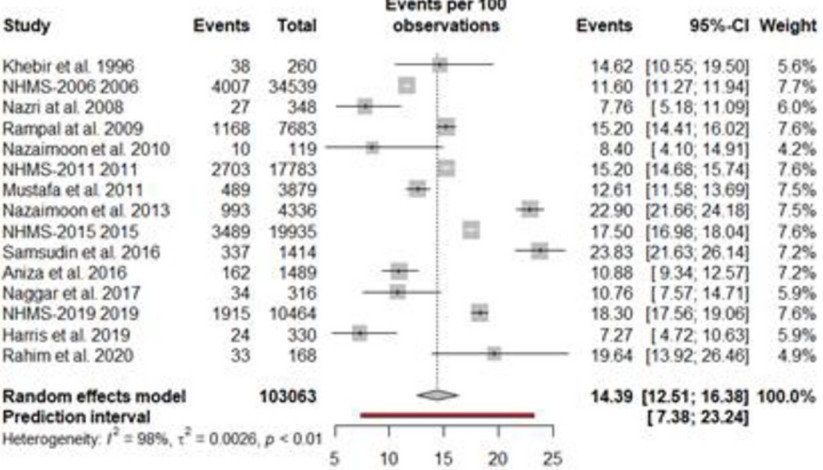

**Fig 2. Pooled prevalence of type-2 diabetes in Malaysia.**

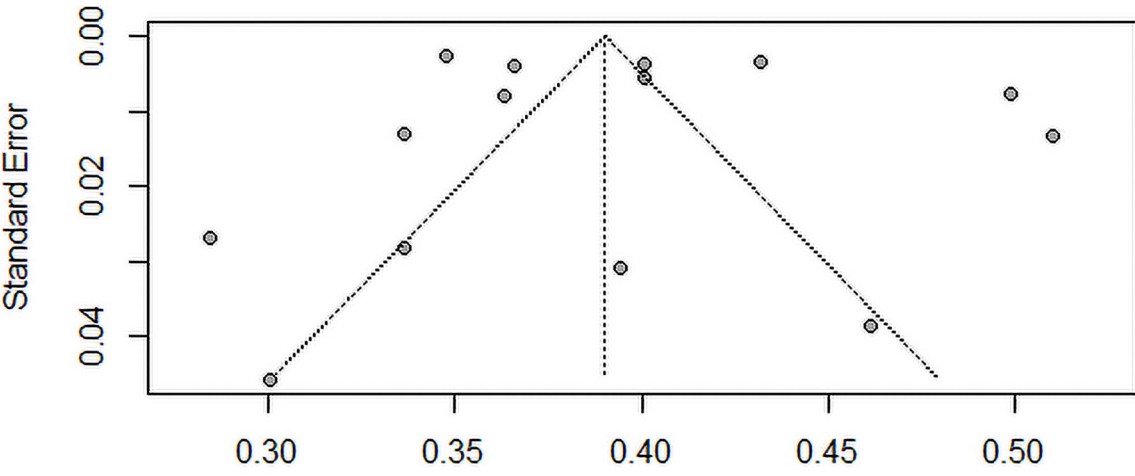

**Fig 3. Funnel plot of the prevalence of type-2 diabetes in Malaysia.**

According to GRADE approach (Table 3), the quality of evidence on the pooled prevalence of diabetes was found to be of low certainty because of inconsistency (the presence of high heterogeneity).

Table 2 also demonstrates the prevalence of diabetes according to gender, setting (rural or urban), ethical sub-groups, time period and age of participants. The prevalence of prediabetes and diabetes did not differ significantly when stratified by sex. The pooled prevalence of diabetes by gender was 13.80% (95% CI, 11.94%–15.77%; $I^2$ = 96.1%) for men and 14.54% (95% CI, 12.50%–16.70%; $I^2$ = 97.3%) for women in the population-based studies while pooled

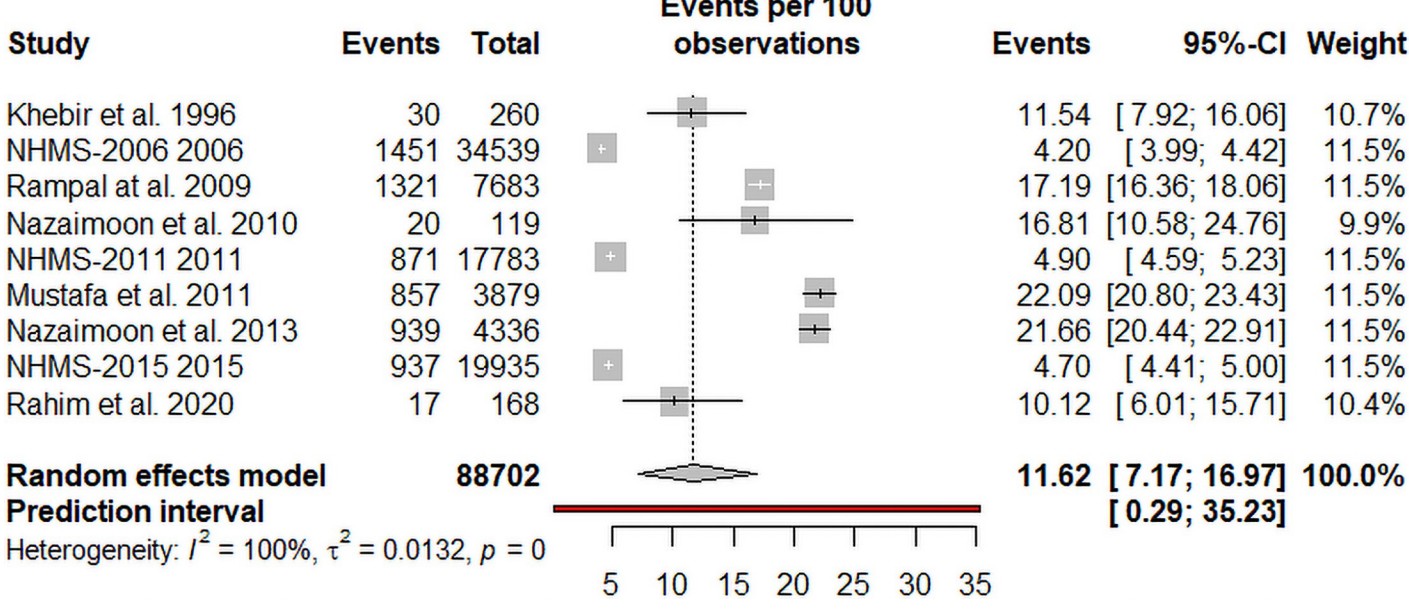

**Fig 4. Pooled prevalence of prediabetes in Malaysia.**

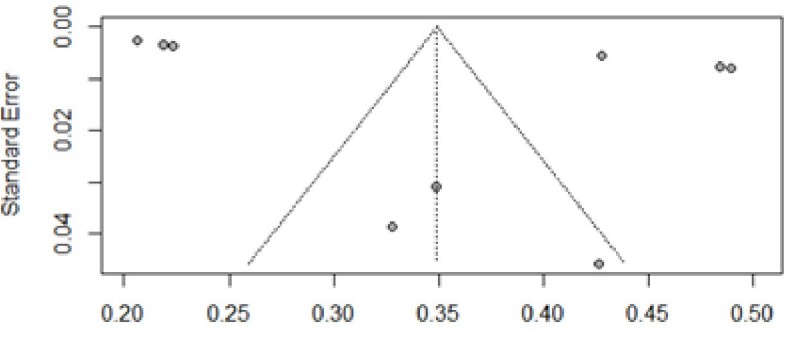

**Fig 5. Funnel plot of the prevalence of prediabetes in Malaysia.**

prevalence of prediabetes was 11.40% (95% CI, 6.55%–17.40%; $I^2$ = 99.7%) for women and 10.98% (95% CI, 6.95%–15.79%; $I^2$ = 99.4%) for men. The prevalence of diabetes for each ethical subpopulation was highest in the Indians sub-population (25.10%; 95% CI, 20.19%–30.35%), followed by Malays (15.25%; 95% CI, 11.59%–19.29%), Chinese (12.87%; 95% CI, 9.73%–16.37%), Bumiputeras (8.62%; 95% CI, 5.41%–12.47%) and others (6.91%; 95% CI, 5.71%–8.19%). There was a significant (p<0.001) increasing trend in diabetes prevalence with increasing age, from 3.16% (95% CI: 3.62%–6.94%) in the 20–29 years age group, 13.71% (95% CI, 12.85%–14.60%) in the 30–45 years age group, 25.66% (95% CI, 20.60%–31.07%) in the 46–59 years of age group, and 33.45% (95% CI, 28.45%–38.64%) in the 60 and above years age groups.

Diabetes prevalence stratified by publication period: 1995–2010, and 2011–2020. Diabetes prevalence was 11.82% (95% CI, 9.44%–14.43%) and 15.77% (95% CI, 13.75%–17.89%) for the publication periods, respectively. Over a 26-year period (1995–2020), the pooled prevalence of diabetes raised from 11.82% to 15.77%. Furthermore, there was no significant difference in the prevalence of diabetes based on diagnostic method (FBG/2hBG, 14.95%; FBG, 10.32%, with p = 0.2058). For all subgroup analyses, there was no publication bias.

Random-effects univariable meta-regression models (Table 4) revealed a stronger association with prevalence of diabetes and increasing age year (β = 0.0073; 95% CI: 0.0021–0.0125, p = 0.0059; $R^2$ = 12.45%), study year (β = 0.0039; 95% CI: 0.0007–0.0071, p = 0.0173; $R^2$ =

**Table 3. GRADE assessment for the studies included in the synthesis with meta-analysis.**

| Nº of studies | Quality assessment | | | | | | Estimated effect (95% CI) | Quality of Evidence |
|---|---|---|---|---|---|---|---|---|
| | Study design | Participants | Risk of Bias | Inconsistency | Indirectness | Imprecision | Publication Bias | |
| 15 | Observational studies | 103063 | Serious[1] | Very Serious[2] | Serious[3] | Serious[4] | Not Serious[5] | 14.39% (12.51–16.38) | ⊕⊕ LOW |

[1] Study quality (assessed by the JBI Critical Appraisal Checklist for Studies Reporting Prevalence Data) ranged from high to low risk of bias. More than 50% of studies included in this analysis were of low risk of bias.

[2] Based on substantial heterogeneity ($I^2$ ranged from 0.00% to 99·8%) and differing estimates of the effect across studies.

[3] Downgrade for serious indirectness due age and ethnical variation, therefore affecting the generalizability to the general population.

[4] The spread of the 95% CI exceeded 10% (±5%); and Only 2 studies had large 95% CIs.

[5] Visual inspection of funnel plot found no evidence of publication bias. The Egger's and Begg's tests also did not find any publication bias in the meta-analysis (p = 0.7296 and p = 0.5862, respectively).

**Table 4. Summary of findings univariate meta-regression analyses.**

| Variable | Beta (*β*) | p-value | 95%CI | $R^2$% |
|---|---|---|---|---|
| Publication year | 0.0025 | 0.1975 | (-0.0013–0.0063) | 44.27 |
| Study year | 0.0039 | 0.0173 | (0.0007–0.0071) | 53.13 |
| Age | 0.0073 | 0.0059 | (0.0021–0.0125) | 12.45 |
| Hypertension | 0.0004 | 0.5777 | (-0.0011–0.0020) | 4.60 |
| Methodology | -0.0387 | 0.1325 | (-0.0898–0.0123) | 14.89 |
| Overweight | -0.0042 | 0.1484 | (-0.0100–0.0015) | 40.69 |
| Gender (male) | 0.0014 | 0.3164 | (-0.0013–0.0040) | 0.00 |
| Setting | -0.1141 | 0.0438 | (-0.2250–-0.0032) | 12.63 |
| Smoking | -0.0015 | 0.4588 | (-0.0056–0.0025) | 0.00 |

53.13%), and study setting (β = -0.1141; 95% CI: -0.2250–-0.0032, p = 0.0438; $R^2$ = 12.63%). The diabetes prevalence was not statistically associated with gender, overweight, smoking status, and quality of studies.

## Discussion

The primary goal of this systematic review and meta-analysis was to collect all available data on the prevalence of diabetes and prediabetes, as well as the risk factors associated with them, among adults in Malaysia between 1995 and 2021. The findings of this study will aid in the development of public health strategies to reduce the prevalence of prediabetes and diabetes. This analysis included 15 studies with a total of 103063 participants. The pooled prevalence of diabetes was 14.39%–approximately one out of every 7 people living in Malaysia is suffering from diabetes. Our results are consistent with a meta-analysis examining the prevalence of diabetes in another Asian country (Pakistan 14.62%) [20]. However, the prevalence of diabetes in Malaysia is significantly higher than the neighboring countries, like Singapore (5.5%) [39] and Indonesia (6.2%) [40].

When age groups were compared, the prevalence of diabetes in the 20–29 year age group was the lowest (3.16%), while the highest prevalence was reported in the 60 and older age group (33.46%). People aged 60 and more are more than 10 times as likely as those aged 20–29 to have type-2 diabetes. This is because that Malaysians is a fast-ageing population in world with average age is 74.4 years in 2017 [41,42]. At the population level, the increasing ageing population and low death rates will increase the proportion of people living with diabetes, putting a large number of people at risk of acquiring sequelae [43].

By stratifying ethically, subgroup analysis revealed that there is a strong association between pooled diabetes prevalence and major ethnic groups. The prevalence of diabetes was most common in the Indian's subpopulation (25.10%), followed by the Malays (15.25%), Chinese (12.87%), Bumiputera (8.62%) and others (6.91%). The results also suggest that the prevalence of diabetes is not influenced by year of publication, sex distribution, gender ratio, methodological quality of studies, setting and smoking.

This study has certain limitations. As expected, we found significant heterogeneity in the included studies, suggesting that about 97% of the variability in the measure of the prevalence of diabetes is due to heterogeneity between studies rather than chance. Subgroup analysis and meta-regression models were used to address the issue of high heterogeneity, with factors added to the univariate model. The findings of this meta-analysis should be interpreted with caution due to the high degree of heterogeneity. Another shortcoming of this review is that the articles chosen did not differentiate between type 1 and type 2 diabetes. As a result, we

hypothesized that all cases of diabetes reported were type-2 which accounts for 90 to 95 percent of all diabetes cases. Furthermore, this study is based on only a few publications (only 15). Because of the small number of studies included in this review, only univariate meta-regression (instead of multivariable meta-regression) analysis is used to analyze the importance of each covariate.

Despite these limitations, this is the first systematic review and meta-analysis to provide pooled prevalence of diabetes and prediabetes in Malaysia. Before beginning the study, we published a protocol outlining our methodology, and we employed scientific and statistical procedures to collect and pool data. Subgroup studies and random effect meta-regression analyses were performed to evaluate the numerous factors that could affect our estimate.

## Conclusion

This study comprehensively describes the prevalence of prediabetes and diabetes in adult population from Malaysia from 1995 to 2021. This study suggests that the pooled diabetes prevalence in Malaysia was 14.39%, but varied significantly by ethical subpopulations with the highest in Indians and lowest in Bumiputeras. The prevalence of diabetes is higher than in neighbouring countries such as Indonesia and Singapore.

Because diabetes and prediabetes are on the increase in Malaysia, the Malaysian government should establish diabetes control programmes throughout the country. To minimize the prevalence of diabetes in Malaysia, the Malaysian government should develop a comprehensive approach and strategy to enhance diabetes awareness, control, prevention, and treatment.

## Supporting information

**S1 Fig. Forest plot of the sensitivity analysis for prevalence of type-2 diabetes in Malaysia.**
(TIF)

**S2 Fig. Forest plot of the sensitivity analysis for prevalence of prediabetes in Malaysia.**
(TIF)

**S1 Table. Search strategies for electronic databases.**
(DOCX)

**S1 File. PRISMA checklist.**
(DOC)

## Author Contributions

**Conceptualization:** Sohail Akhtar.

**Data curation:** Sohail Akhtar, Jamal Abdul Nasir, Aqsa Ali, Mubeen Asghar, Rizwana Majeed.

**Formal analysis:** Sohail Akhtar, Mubeen Asghar.

**Investigation:** Sohail Akhtar, Aqsa Ali, Mubeen Asghar, Rizwana Majeed.

**Methodology:** Sohail Akhtar.

**Software:** Sohail Akhtar, Aqsa Ali, Rizwana Majeed.

**Supervision:** Sohail Akhtar, Jamal Abdul Nasir.

**Validation:** Rizwana Majeed.

**Visualization:** Sohail Akhtar.

Writing – **original draft:** Sohail Akhtar.

Writing – **review & editing:** Jamal Abdul Nasir, Aqsa Sarwar.

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
