## [Decision Letter · Decision Letter 0]

15 Nov 2021

PONE-D-21-32696Prevalence of diabetes and prediabetes in Malaysia: a systematic review and meta-analysisPLOS ONE

Dear Dr. Akhtar,

Thank you for submitting your manuscript to PLOS ONE. After careful consideration, we feel that it has merit but does not fully meet PLOS ONE’s publication criteria as it currently stands. Therefore, we invite you to submit a revised version of the manuscript that addresses the points raised during the review process.

Please read carefully the reviewers' comments and improve your manuscript as for their suggestions.

We look forward to receiving your revised manuscript.

Kind regards,

Giulio Francesco Romiti

Academic Editor

PLOS ONE

Journal Requirements:

"The funders had no role in study design, data collection and analysis, decision to publish, or preparation of the manuscript"

Reviewers' comments:

Reviewer's Responses to Questions

5. Review Comments to the Author

Reviewer #1: Thank you for giving me the opportunity to review this manuscript. The authors conducted a systematic review and meta-analysis to investigate the pooled prevalence of prediabetes and diabetes in the general population of Malaysia. I have few comments that I feel authors should address before the manuscript is considered for publication.

1.The introduction part is not sufficiently described on the problem statement of diabetes in Malaysia. It lacks motivation to conduct the current study. I believe authors have not elaborated the burden of diabetes and that the studies conducted in Malaysia could apprehend the true “burden” of diabetes and pre-diabetes based on established works till date. The following literature that was conducted from the Malaysian perspective needs to be critically appraised, elaborated and cited in the current paper:

- Ganasegeran K, Hor CP, Jamil MFA, Loh HC, Noor JM, Hamid NA, Suppiah PD, Abdul Manaf MR, Ch’ng ASH, Looi I. A Systematic Review of the Economic Burden of Type 2 Diabetes in Malaysia. International Journal of Environmental Research and Public Health. 2020; 17(16):5723. https://doi.org/10.3390/ijerph17165723

- Ganasegeran K, Hor CP, Jamil MFA, Suppiah PD, Noor JM, Hamid NA, Chuan DR, Manaf MRA, Ch’ng ASH, Looi I. Mapping the Scientific Landscape of Diabetes Research in Malaysia (2000–2018): A Systematic Scientometrics Study. International Journal of Environmental Research and Public Health. 2021; 18(1):318. https://doi.org/10.3390/ijerph18010318

2. In addition, the authors could highlight the burden of diabetes of neighboring countries, especially within the Western Pacific region, in comparison to Malaysia.

3. In the PRISMA chart, what do the authors meant by “additional records identified through other sources?” What are the sources were used? This needs to be clarified.

4. In the PRISMA chart again, I could not capture authors intention of excluding up to 307 full text articles with reason. What reasons?

5. Please attach the results of your search strategies for each database used as your supplementary file.

6. Classifying studies as high risks bias, moderate risk bias and low risks from the methodological quality assessment seems a little awkward. Cross-sectional studies are often not powered to establish causality or temporality, instead mere associations and they can be self-reported. Were these taken into consideration?

7. Is the tool for methodological assessment appropriate? Authors mentioned four studies did not explicitly mention study design. Then why was it included and how could it be interpreted to extract prevalence rates?

8. Most studies are single-centered. Will they be sufficiently be powered to be pooled for a country level estimate?

9. Authors included the NHMS surveys as well. Kindly note that NHMS is the nation’s five year survey of exploring health of the Malaysian people. It is a report published and not those studies published as reported in the methods part (extracted from databases) as claimed by the authors.

10. Inclusion criteria is not clear on the type pf diabetes – type 1, type 2, gestational or all?

Authors need to undertake the revisions as suggested above, to be further considered for publication.

Reviewer #2: In this paper “Prevalence of diabetes and prediabetes in Malaysia: a systematic review and meta-analysis”, the authors (Sohail Akhtar and et al) estimated the prevalence of diabetes and pre-diabetes in Malaysia population. To do so, they reviewed published articles until September 30, 2021 and then pooled all values of prevalence. Unfortunately, the manuscript contained many flaws, some were serious:

Title & Introduction section

- Type of diabetes must be specified.

- References (e.g. number 27, 29, 31, and … must be written in Vancouver reference style.

- Introduction should be short, summary, and comprehensive.

Method section

- Search strategy is not well-defined. Authors must apply some key words such as ‘prediabetes’ , ‘gestational diabetes’

- Please describe about reasons of excluding 307 studies.

- Exclusion and inclusion criteria should be reported for 307 excluded studies.

- Gestational diabetes cases must be categorized separately.

Results

- In results section, first paragraph must describe first diagram (e.g. relevant studies=54).

- In second paragraph, description about gender frequency has been repeated.

- Please report number of studies by type of diabetes(type I and type II)

- Please report prevalence by diagnostic method

- In table 1, please specify positives for diabetes, pre-diabetes, …

- In table 1, it seems that some subjects are wrong. E.g. the prevalence for Smsudin et al (14) study in original study is 69 but in present study reported 65. Also, in original full text, snowball sampling (non-random) is method of sampling but authors reported ‘Random’ in Table 1.

- In Table1, citing of studies is incorrect. Please check it.

- In section of ‘quantitative synthesis’, prevalence for men reported 14.63 but it is 16.15 in table 2.

- Table 2 has some typing errors.

- In table 3, please specify type of gender.

- The authors must check all extracted data.

- The statistical analysis is not good and not representing the results properly.

- Non-significant variables in univariate analysis did not discuss in discussion section.

- Authors did not carry out Sensitivity analysis.

Reviewer #3: 1. GRADE scoring was not used to assess the quality of meta-analysis evidence. Evaluate your meta-analysis using the GRADE score.

2. Diabetes diagnosis criteria were not stated for studies, as it was changed over the years.

3. Studies reporting prevalence may not be published or can be published in local databases. Your search must be amended by searching for these publications and gray literature.

4. Article screening process was well described, but the numbers stated are wrong.

5. In the abstract of the article and the results, two different population sizes are mentioned. It should be corrected.

6. In the result section “Male participation ranged from 0 to 61.8%, and the number of overweight participants ranged from 11 percent to 49.1 percent". The "number" should be changed to "percentage".

7. In the result section “table 1 summarize……”. The study is not a longitudinal study. Thus using baseline characteristics is inappropriate, it should be changed to general characteristics.

---

## [Author Response · Author response to Decision Letter 0]

10 Dec 2021

PONE-D-21-32696

Prevalence of diabetes and prediabetes in Malaysia: a systematic review and meta-analysis

PLOS ONE

Dear Dr. Akhtar,

Thank you for submitting your manuscript to PLOS ONE. After careful consideration, we feel that it has merit but does not fully meet PLOS ONE’s publication criteria as it currently stands. Therefore, we invite you to submit a revised version of the manuscript that addresses the points raised during the review process.

Reply: Thanks a lot for your response and comments. All the comments of the reviews have been carefully considered and incorporated in the revised version. The point-by-point response is given below here and mentioned in the revised version in highlighted color, as given below. 

Response to Reviewer 1: All changes have been mentioned in Yellow Colour

Response to Reviewer 2: All changes have been mentioned in Green Colour

Response to Reviewer 3: All changes have been mentioned in Pink Colour

Please note that:

Answer: We don’t have any sources of funding for this study.

Answer: The funders had no role in study design, data collection and analysis, decision to publish, or preparation of the manuscript

Answer: No author received any salary form any funder.

d) If you did not receive any funding for this study,

 Answer:The authors received no specific funding for this work

Reviewer #1:

Reviewer #1: Thank you for giving me the opportunity to review this manuscript. The authors conducted a systematic review and meta-analysis to investigate the pooled prevalence of prediabetes and diabetes in the general population of Malaysia. I have few comments that I feel authors should address before the manuscript is considered for publication.

1.The introduction part is not sufficiently described on the problem statement of diabetes in Malaysia. It lacks motivation to conduct the current study. I believe authors have not elaborated the burden of diabetes and that the studies conducted in Malaysia could apprehend the true “burden” of diabetes and pre-diabetes based on established works till date. The following literature that was conducted from the Malaysian perspective needs to be critically appraised, elaborated and cited in the current paper:

- Ganasegeran K, Hor CP, Jamil MFA, Loh HC, Noor JM, Hamid NA, Suppiah PD, Abdul Manaf MR, Ch’ng ASH, Looi I. A Systematic Review of the Economic Burden of Type 2 Diabetes in Malaysia. International Journal of Environmental Research and Public Health. 2020; 17(16):5723. https://doi.org/10.3390/ijerph17165723

- Ganasegeran K, Hor CP, Jamil MFA, Suppiah PD, Noor JM, Hamid NA, Chuan DR, Manaf MRA, Ch’ng ASH, Looi I. Mapping the Scientific Landscape of Diabetes Research in Malaysia (2000–2018): A Systematic Scientometrics Study. International Journal of Environmental Research and Public Health. 2021; 18(1):318. https://doi.org/10.3390/ijerph18010318

Answer: Thanks a lot for your comments. Both the references have been added in the introduction section (see the yellow highlighted section, page1 (2nd paragraph).

2. In addition, the authors could highlight the burden of diabetes of neighboring countries, especially within the Western Pacific region, in comparison to Malaysia.

Answer: A comparison has been added. See highlighted in yellow in introduction section, page 1

3. In the PRISMA chart, what do the authors meant by “additional records identified through other sources?” What are the sources were used? This needs to be clarified.

Answer: Additional records identified through hand searching of reference list of selected articles. Corrected accordingly on page 12 and already mentioned in search strategy. 

4. In the PRISMA chart again, I could not capture authors intention of excluding up to 307 full text articles with reason. What reasons?

Answer: Irrelevant articles were excluded on the basis of title and abstract review. Corrected accordingly in the PRISMA chart on page 12 (highlighted in yellow)

5. Please attach the results of your search strategies for each database used as your supplementary file.

Answer: Thanks a lot for your comment. Added on supplementary life.

6. Classifying studies as high risks bias, moderate risk bias and low risks from the methodological quality assessment seems a little awkward. Cross-sectional studies are often not powered to establish causality or temporality, instead mere associations and they can be self-reported. Were these taken into consideration?

Answer: Critical appraisal is a systematic process used to identify the strengths and weaknesses of a research article to assess the usefulness and validity of research findings. The most important components of a critical appraisal are an evaluation of the appropriateness of the study design for the research question and a careful assessment of the key methodological features of this design. 

We have now considered another appraisal for prevalence study“JBI Critical Appraisal Checklist for Studies Reporting Prevalence Data”. The purpose of this appraisal is to evaluate the methodological quality of a study and to determine the extent to which a study has addressed the possibility of bias in its design, conduct and analysis

7. Is the tool for methodological assessment appropriate? Authors mentioned four studies did not explicitly mention study design. Then why was it included and how could it be interpreted to extract prevalence rates?

Answer: Thanks a lot for your comment. The four studies which did not explicitly mention study design but can be inferred as a cross-section study design (as mentioned now result section. However, we have changed the critical appraisal from “Quality Assessment Tool for Observational Cohort and Cross-Sectional Studies” to “JBI Critical Appraisal Checklist for Studies Reporting Prevalence Data” to check the methodological quality of included articles.

8. Most studies are single-centered. Will they sufficiently be powered to be pooled for a country level estimate?

Asnwer: Meta-analysis is a research process used to systematically synthesise or merge the findings of single (single or multi-centered), independent studies, using statistical methods to calculate an overall or ‘absolute’ effect.

9. Authors included the NHMS surveys as well. Kindly note that NHMS is the nation’s five-year survey of exploring health of the Malaysian people. It is a report published and not those studies published as reported in the methods part (extracted from databases) as claimed by the authors.

Answer: We also scrutinized the reference list for additional studies and reports, as mentioned in the literature search section. The NHMS surveys were found in reference lists from different articles.

10. Inclusion criteria is not clear on the type of diabetes – type 1, type 2, gestational or all?

Answer: Articles only on type-2 diabetes were considered for this systematic review and meta-analysis. The title and inclusion exclusion criteria have been updated accordingly. 

Authors need to undertake the revisions as suggested above, to be further considered for publication.

Reviewer #2:

In this paper “Prevalence of diabetes and prediabetes in Malaysia: a systematic review and meta-analysis”, the authors (Sohail Akhtar and et al) estimated the prevalence of diabetes and pre-diabetes in Malaysia population. To do so, they reviewed published articles until September 30, 2021 and then pooled all values of prevalence. Unfortunately, the manuscript contained many flaws, some were serious:

Title & Introduction section

- Type of diabetes must be specified.

Answer: Thanks a lot for your comment. Title and paper are modified accordingly.

- References (e.g., number 27, 29, 31, and … must be written in Vancouver reference style.

Answer: Checked and changed to Vancouver reference style accordingly.

- Introduction should be short, summary, and comprehensive.

Answer: Introduction section is reduced now in comprehensive way.

Method section

- Search strategy is not well-defined. Authors must apply some key words such as ‘prediabetes’, ‘gestational diabetes’

Answer: The term ‘prediabetes’ is added now search strategy, and “gestational diabetes” is added in the exclusion criteria. We only added articles on type-2 diabetes.

- Please describe about reasons of excluding 307 studies.

Answer: Irrelevant articles excluded based on title and abstract review (n = 733). Corrected accordingly on page 12 (PRISMA Chart).

- Exclusion and inclusion criteria should be reported for 307 excluded studies

Answer: The following exclusion criteria is present, section “Inclusion and exclusion criteria” on page 3 (highlighted in green). “The following studies were excluded: were review articles, case studies, qualitative studies, case series, abstracts, and intervention studies; was irrelevant to type-2 diabetes; reported on gestational or type-1 diabetes”.

- Gestational diabetes cases must be categorized separately.

Answer: We have only considered type-2 diabetes in this systematic review and meta-analysis. Therefore, gestational diabetes is added in exclusion criteria and type-2 diabetes is mentioned in the title now. See, page 3 (highlighted in green)

Results

- In results section, first paragraph must describe first diagram (e.g. relevant studies=54).

Answer: First diagram is described now, page 5.

- In second paragraph, description about gender frequency has been repeated.

Answer: Gender repetition has been deleted now.

 - Please report number of studies by type of diabetes (type I and type II)

Answer: We have only considered type-2 diabetes in this systematic review and meta-analysis. 

- Please report prevalence by diagnostic method

Answer: Most of the studies did not report the prevalence of diabetes by diagnostic methods. The prevalence of diagnostic method is presented on page number 8 (highlighted in green).

- In table 1, please specify positives for diabetes, pre-diabetes, …

Answer: The positive of diabetes and prediabetes are now described in table 1 (highlighted in green)

- In table 1, it seems that some subjects are wrong. E.g. the prevalence for Smsudin et al (14) study in original study is 69 but in present study reported 65. Also, in original full text, snowball sampling (non-random) is method of sampling, but authors reported ‘Random’ in Table 1.

Answer: Thanks a lot for your comment. Firstly, age value is corrected now in the table 1. Secondly, stratified sampling was applied for the selection of sub-areas and later they used snowball sample to choose the housing estates. Therefore, both (Stratified random and Snowball sampling are mentioned in table 1)

- In Table1, citing of studies is incorrect. Please check it.

Answer Checked and corrected accordingly.

- In section of ‘quantitative synthesis’, prevalence for men reported 14.63 but it is 16.15 in table 2.

Answer Checked and corrected accordingly.

- Table 2 has some typing errors.

Answer Re-checked and corrected accordingly.

- In table 3, please specify type of gender.

Answer Done accordingly.

- The authors must check all extracted data.

Answer The extracted data is rechecked and corrected accordingly.

- The statistical analysis is not good and not representing the results properly.

Answer Statistical analysis has significantly improved by adding Sensitive Analysis, Grade approach, Begg’s test, etc (highlighted in green)

- Non-significant variables in univariate analysis did not discuss in discussion section.

Answer Added in the discussion section.

- Authors did not carry out Sensitivity analysis.

Answer Sensitivity analysis has been added now.

Reviewer #3:

1. GRADE scoring was not used to assess the quality of meta-analysis evidence. Evaluate your meta-analysis using the GRADE score.

Answer: Thanks a lot for your suggestion. GRADE scoring has been added accordingly. Mentioned in result section on page 9.

2. Diabetes diagnosis criteria were not stated for studies, as it was changed over the years.

Answer: Mentioned now accordingly in table 1.

3. Studies reporting prevalence may not be published or can be published in local databases. Your search must be amended by searching for these publications and gray literature.

Answer: Thanks a lot for your comment. Malaysia online journals are searched too now. We have founded one more relevant through local journals.

4. Article screening process was well described, but the numbers stated are wrong.

Answer: Corrected accordingly 

5. In the abstract of the article and the results, two different population sizes are mentioned. It should be corrected.

Answer: Thanks a lot for your comment. Corrected accordingly

6. In the result section “Male participation ranged from 0 to 61.8%, and the number of overweight participants ranged from 11 percent to 49.1 percent". The "number" should be changed to "percentage".

Answer: Thanks a lot for your correction. Changed it accordingly

7. In the result section “table 1 summarize……”. The study is not a longitudinal study. Thus using baseline characteristics is inappropriate, it should be changed to general characteristics.

Answer: Changed it accordingly

---

## [Decision Letter · Decision Letter 1]

13 Jan 2022

Prevalence of type-2 diabetes and prediabetes in Malaysia: a systematic review and meta-analysis

PONE-D-21-32696R1

Dear Dr. Akhtar,

We’re pleased to inform you that your manuscript has been judged scientifically suitable for publication and will be formally accepted for publication once it meets all outstanding technical requirements.

Kind regards,

Giulio Francesco Romiti

Academic Editor

PLOS ONE

Additional Editor Comments (optional):

Reviewers' comments:

Reviewer's Responses to Questions

6. Review Comments to the Author

Reviewer #1: Thank you for addressing my previous comments. The responses to my previous comments have been addressed and justified appropriately. The revisions are satisfactory.

Reviewer #3: Significant changes have been made from the original submission. I believe the article should be considered for acceptance.

---

## [Editor Report · Acceptance letter]

17 Jan 2022

PONE-D-21-32696R1 

Prevalence of type-2 diabetes and prediabetes in Malaysia: a systematic review and meta-analysis 

Dear Dr. Akhtar:

I'm pleased to inform you that your manuscript has been deemed suitable for publication in PLOS ONE. Congratulations! Your manuscript is now with our production department. 

Kind regards, 

on behalf of

Dr. Giulio Francesco Romiti 

Academic Editor

PLOS ONE